# Synthesis of Tungsten-Doped Vanadium Dioxide Using a Modified Polyol Method Involving 1-Dodecanol

**DOI:** 10.3390/ma13235384

**Published:** 2020-11-27

**Authors:** Yonghyun Lee, Sang Won Jung, Sang Hwi Park, Jung Whan Yoo, Juhyun Park

**Affiliations:** 1Department of Intelligent Energy and Industry, School of Chemical Engineering and Materials Science, Chung-Ang University, Seoul 06974, Korea; lovembnb@cau.ac.kr (Y.L.); jung11361@naver.com (S.W.J.); 2KNW R&D Center, Gyeonggi-do 10832, Korea; kunchs@knwkorea.com (S.H.P.); jwyoo@knwkorea.com (J.W.Y.)

**Keywords:** vanadium oxide, thermochromic, nanoparticles, tungsten doping, smart windows

## Abstract

The doping of tungsten into VO_2_ (M) via a polyol process that is based on oligomerization of ammonium metavanadate and ethylene glycol (EG) to synthesize a vanadyl ethylene glycolate (VEG) followed by postcalcination was carried out by simply adding 1-dodecanol and the tungsten source tungstenoxytetrachloride (WOCl_4_). Tungsten-doped VEGs (W-VEGs) and their calcinated compounds (W_x_VO_2_) were prepared with varying mixing ratios of EG to 1-dodecanol and WOCl_4_ concentrations. Characterizations of W-VEGs by powder X-ray diffraction, differential scanning calorimetry, scanning electron microscopy, and infrared and transmittance spectroscopy showed that tungsten elements were successfully doped into W_x_VO_2_, thereby decreasing the metal-insulator transition temperature from 68 down to 51 °C. Our results suggested that WOCl_4_ variously combined with 1-dodecanol might interrupt the linear growth of W-VEGs, but that such an interruption might be alleviated at the optimal 1:1 mixing ratio of EG to 1-dodecanol, resulting in the successful W doping. The difference in the solar modulations of a W_0.0207_VO_2_ dispersion measured at 20 and 70 °C was increased to 21.8% while that of a pure VO_2_ dispersion was 2.5%. It was suggested that WOCl_4_ coupled with both EG and 1-dodecanol at an optimal mixing ratio could improve the formation of W-VEG and W_x_VO_2_ and that the bulky dodecyl chains might act as defects to decrease crystallinity.

## 1. Introduction

Smart windows that use the thermochromic property of vanadium dioxide (VO_2_) have been highlighted as a promising energy saving technology due to their selective control over the transmission of heat rays that enhances energy saving efficiency [1]. At 68 °C or below, VO_2_ has a monoclinic structure (M) with a semiconducting property, and at temperatures above 68 °C, its crystal structure is transformed to a tetragonal rutile structure, accompanying a change in its electrical property from semiconducting to metallic [2]. This metal-insulator transition (MIT) enables VO_2_ to selectively reflect near-infrared light (NIR) due to its metallic property above its MIT temperature while transmitting visible light [3]. To practically apply the thermochromic property of VO_2_ to smart windows that can control sunlight transmission at room temperature, decreasing the MIT temperature of VO_2_ down to around room temperature is very much required. The doping of VO_2_ (M) with tungsten (W) atoms has been a promising route for reducing the MIT temperature of VO_2_ (M) [4,5,6,7]. When tungsten atoms are doped to the VO_2_, it is expected that the original V^4+^-V^4+^ pairs in VO_2_ (M) are converted to V^3+^-V^4+^ and V^3+^-W^6+^ pairs by electron donation from W to the neighboring V ions. As a result, the concentration of free electrons is increased via electron donation, and the monoclinic structure is distorted due to the insertion of larger doped W^6+^ ions with an ionic radius of 64 pm than V^4+^ ions with an ionic radius of 49.5 pm. Thus, the MIT temperature of pristine VO_2_ is decreased, making W_x_VO_2_ (M) an ideal candidate for thermochromic window materials as it can reflect NIR at room temperature while transmitting visible light [8,9].

A variety of methods, including hydrothermal synthesis with an autoclave at a high pressure [10], chemical vapor deposition (CVD) [11], sputtering [12,13], and ion implantation [14], have been utilized to synthesize W_x_VO_2_ (M). However, these methods require complicated experimental parameter controls or special devices, thereby restricting the extension of these methods to large-scale production at a low cost. Thus, it is highly necessary to develop a convenient process that utilizes atmospheric pressure in air at a low processing temperature. A candidate method for resolving these issues is the thermolysis of vanadyl glycolate to synthesize VO_2_ (M) [15,16]. In this method, an alcohol with two or more hydroxyl groups and high boiling points, such as ethylene glycol (EG) with a boiling point of 197 °C, is mixed with a cheap vanadium precursor, such as ammonium metavanadate (NH_4_VO_3_), and the resulting solution mixture is heated over 160 °C in atmospheric air followed by thermolysis of precipitates at a low pyrolytic temperature of 200 °C to produce VO_2_ (M) (Scheme 1). In this polyol method, EG works as a stabilizer that limits particle growth and prevents aggregation [17], as a coupling agent that can further lead vanadyl ethyleneglycol glycolate (VEG) to oligomer with a repeat unit of VO(OCH_2_CH_2_O) [18], and as a reducing agent when the VEG is calcinated [19]. The crystal structure of VEG can also be depicted by the one-dimensional chain of the VO_5_ square pyramids in the VO(OCH_2_CH_2_O) structure by edge sharing (Scheme 1) [19]. When using this method to make VEG, special devices are not required, and subsequent calcination can be carried out via a conventional sintering process, thereby suggesting its possibility for mass production at a low cost. Although the polyol method shows great potential for the facile synthesis of W_x_VO_2_, it is difficult to find reports about the polyol method being applied to synthesize W_x_VO_2_ nanoparticles. This lack of research seems to be because a special synthesis for tungsten precursors that can be coupled within VEG is necessary or because the insertion of the large W^6+^ ions into the monoclinic VO_2_ (M) at an atmospheric pressure is difficult. Thus, the synthesis of W_x_VO_2_ using the polyol method via thermolysis in atmospheric air remains challenging.

In this study, we present a modified polyol method using 1-dodecanol as an additive to synthesize W_x_VO_2_ (M). 1-Dodecanol has a boiling point of 259 °C, which is high enough for the polyol method, and can act as both the solvent and capping agent, as shown in inorganic nanoparticle synthesis [20]. Unlike the original polyol method, we found that tungsten could be doped into VO_2_ (M) with the addition of 1-dodecanol, thereby reducing the MIT temperature of the resulting W_x_VO_2_ (M). We investigated the influence of the mixing ratios of EG and 1-dodecanol and the amounts of tungsten precursor on the synthesis of W_x_VO_2_. Our results suggest a procedure to dope tungsten into VO_2_ (M) via the polyol method.

## 2. Materials and Methods

### 2.1. Materials

NH_4_VO_3_ and tungsten(VI) oxytetrachloride (WOCl_4_) were purchased from Sigma-Aldrich Co. (St. Louis, MO, USA). EG and 1-dodecanol were purchased from Alfa Aesar (Lancashire, United Kingdom). Ethanol was obtained from SamChun Pure Chemicals Co. (Seoul, South Korea). All the reagents used in this study were used as received.

### 2.2. Synthesis

To first synthesize tungsten-doped VEG (W-VEG), NH_4_VO_3_ (1.17 g, 10 mmol), 1-dodecanol (33.17 g, 17.8 mmol), and EG (11.05 g, 17.8 mmol) were added to a three-necked round-bottom flask and mixed with stirring. To this mixture solution, a pre-determined amount of WOCl_4_ (0.094 g [0.277 mmol, 2.77 mol% to NH_4_VO_3_], 0.188 g [0.553 mmol, 5.53 mol%), and 0.377 g [1.109 mmol, 11.08 mol%]) was added, and the resulting reaction mixture was heated to 160 °C at a ramping rate of 5 °C/min and stirred for 2 h. Then, the mixture with a purple color was naturally cooled down to room temperature. The precipitate was collected by centrifuging the solution at 8000 rpm for 15 min to remove the excess 1-dodecanol and EG. The collected purple powder was washed with ethanol, filtered to obtain W-VEG powder, and dried in a vacuum for 24 h at room temperature.

The W-VEG powder was calcinated at 200 °C for 1 h in a box furnace in atmospheric air to become W_x_VO_y_ (M). After cooling to room temperature, W_x_VO_y_ (M) was calcinated again at 700 °C (10 °C/min) for 1 h in a tube furnace in a vacuum to obtain W_x_VO_y_ (M).

### 2.3. Characterization

The crystal structures of the W_x_VO_y_ (M) were verified by an X-ray diffractometer (XRD, Bruker-AXS NEW D8 Advance, Billerica, MA, USA). The MIT behaviors of the W_x_VO_y_ (M) were investigated using a differential scanning calorimeter (DSC Q-600; TA Instruments, New Castle, DE, USA) for analysis in the range −20–130 °C at a heating rate of 10 °C/min. Morphological observations of the nanostructures were conducted on a field-emission scanning electron microscope (FE-SEM; SIGMA, Carl Zeiss, Germany). The thermochromic properties of W_x_VO_y_ (M) were measured using UV-Vis-NIR spectroscopy (V-770, JASCO, Tokyo, Japan) in the range of 300–2500 nm. Before measuring the UV-Vis-NIR spectra, the W_x_VO_y_ (M) powder (0.02 g) was added in 5 mL of ethanol, and the solution was dispersed for 1 h under ultrasonication to make the suspension and placed in a quartz cell with 1 mm light path. The FT-IR spectra were recorded using a high-vacuum FT-IR spectrometer system in a spectral range of 4000 to 400 cm^−1^ (Vertex 80V, operated by the Korea Basic Science Institute at Busan, Bruker Co., Billerica, MA, USA). KBr pellets that were prepared with 1 mg of samples and 15 mg of KBr were used for measurements in a vacuum. The compositions of W_x_VO_y_ were determined by using an Inductively Coupled Plasma Atomic Emission Spectrometer (ICP-AES; Jobin Yvon Ultima2, operated by the Korea Basic Science Institute at Seoul; HORIBA, Edison, NJ, USA).

## 3. Results and Discussion

We used WOCl_4_ as a tungsten precursor to synthesize W_x_VO_2_ (M) via the polyol method based on ammonium metavanadate and EG because WOCl_4_ could be readily combined with EG via an exothermic reaction between WOCl_4_ and hydroxyl groups that make byproducts of HCl. Thus, one might expect that WOCl_4_ combined with EG participates in the formation of VEG, thereby resulting in the insertion of tungsten atoms into VO_2_ (M) during calcination. However, when 2.77 mol% of WOCl_4_ to NH_4_VO_3_ was added into the reaction mixture of ammonium metavanadate and EG at the 1:0 mixing ratio of EG to 1-dodecanol, the final product gained after calcination did not show a highly crystalline structure in its XRD pattern, as shown in Figure 1a. Additionally, the XRD pattern was not matched with that of VO_2_ (M). This result can be ascribed to the incomplete synthesis of linear VEG via a condensation reaction between NH_4_VO_3_ and EG because WOCl_4_ with four reactive W-Cl functionalities to EGs can interrupt the formation of the linear VEG.

We presumed that the simple addition of 1-dodecanol as a co-solvent could resolve this issue because 1-dodecanol might act as a partial capping agent for the highly reactive W-Cl functionality in WOCl_4_. To investigate the capping effect of 1-dodecanol on WOCl_4_, we carried out the synthesis of W_x_VO_2_ by varying the molar mixing ratio of EG to 1-dodecanol at 1:0.5, 1:1, and 1:2 at fixed amounts of NH_4_VO_3_ to WOCl_4_ at 10 and 0.277 mmol, respectively. Figure 1a shows the XRD patterns of the tungsten-doped vanadium oxides produced by controlling the molar ratio of EG:1-dodecanol. The observed diffraction peaks at all three ratios of 1:0.5, 1:1 and 1:2 indicate the monoclinic vanadium dioxide VO_2_ (M) crystal structure reported in JCPDS No. 43-1051 [21]. Particularly, well-developed crystallinity was seen in the main VO_2_(M) peaks at 2θ angles of 26.8°, 27.8° 33.4°, 37.0°, 39.8°, 42.1°, 42.2°, 55.5°, and 57.4° for (110), (011), (−102), (−202; −211; 200), (002), (−212; 210), (−213; 220; 211), and (022) planes, respectively. However, diffraction peaks of V_6_O_13_ at 2θ = 25.3°, 30.1°, and 45.6° for (101), (400), and (005), respectively, were also clearly observed in the XRD patterns of the compounds synthesized at ratios of 1:1 and 1:2, and were in agreement with the reference data (JCPDS No. 25-1251) [22]. The appearance of the oxidized V_6_O_13_ peaks originates from the increase in oxygen contents in W-VEG as the molar ratio of 1-dodecanol increases. It appears that excess oxygen atoms incorporated into the VO_2_ structure chemically interact with vanadium atoms to form a new oxidation state of V_6_O_13_, as stated in literature [23].

The decrease in MIT temperatures with the addition of tungsten precursors in varying molar ratios of EG to 1-dodecanol at the fixed amount of WOCl_4_ to NH_4_VO_3_ was confirmed by measuring the DSC thermogram. In Figure 1b, the DSC curves show that the MIT temperatures were 63.03, 61.23, and 63.44 °C at mixing ratios of 1:0.5, 1:1, and 1:2 for EG to 1-dodecanol. These results show an obvious drop from 68 °C, the original MIT temperature of VO_2_ (M), and indicate successful tungsten doping via the polyol method. Furthermore, it should be noted that W_x_VO_2_ prepared from the 1:1 mixture of EG to 1-dodecanol has the lowest MIT temperature, although the fixed amount of WOCl_4_ was used for all calcinated samples.

To investigate the origination of these variations in the MIT temperatures, we measured SEM images of VEG synthesized without using 1-dodecanol and WOCl_4_ as a reference and W-VEGs prepared with varying mixing ratios of EG to 1-dodecanol at the fixed WOCl_4_ concentration of 2.77 mol% relative to NH_4_VO_3_ (Figure 2). The SEM images of VEG in Figure 2a looks like a mixture of square rods at the nano‒ and micrometer scales, which is similar to nanorods found in the literature [24]. This morphology reflects VEG crystals based on one dimensional chain of square pyramids sharing edges with the formula VO(OCH_2_CH_2_O) [19], as shown in Scheme 1. When 1-dodecanol was not used, the SEM image of W-VEG, as can be seen in Figure 2b, showed microcrystals with rod-like morphologies and a square cross section. The side length of the square cross section was around 1 μm. This morphology did not significantly deviate from that of VEG without using the tungsten precursor, as can be seen in Figure 2a. With the addition of 1-dodecanol at 1:0.5 and 1:2, as shown in Figure 2c,e, the edges of the crystals were smeared, and the resulting morphology was a mixture of rods and smeared aggregates. The SEM image at 1:1 shown in Figure 2d shows morphologies with sharp edges, but the sizes and lengths of the microcrystals were significantly decreased in comparison to those at a 1:0 ratio of 1-dodecanol.

One hypothesis to explain the variation in the morphologies of W-VEG microcrystals is that WOCl_4_ reacted with 1-dodecanol, which significantly influenced the crystal morphologies. The reaction between NH_4_VO_3_ and EG is like the oligomerization of two monomers, resulting in VEG with a one-dimensional chain of VO_5_ square pyramids, as shown in Scheme 1 and the literature [18]. Our results suggest that the WOCl_4_ combined with 1-dodecanol might interrupt the growth of VEG in one dimension, resulting in the decrease in tungsten doping after calcination of the W-VEG, and when at the 1:1 molar mixing ratio, such an interruption might be alleviated. In the mixture of EG and 1-dodecanol, WOCl_4_ can form many different chemical structures shown in Scheme 2. Without using 1-dodecanol, species I, II, and III are primarily formed, and species I might hinder the formation of the one-dimensional chain of the VO_5_ square pyramids due to its multiple functionalities. Species IV, V, VI, and VII, which can be produced by the reaction of WOCl_4_ with both of EG and 1-dodecanol, should also significantly influence the final morphologies of W-VEGs. In particular, species V might be the main product of the reaction between WOCl_4_ and the mixture of EG and 1-dodecanol at a 1:1 ratio due to the statistically equal opportunity for the reaction, which occurs only if the reactivities of WOCl_4_ to EG and 1-dodecanol are identical; this in turn can be responsible for the improved tungsten doping.

To further investigate the credibility of our hypothesis and speculations and the effects of the addition of 1-dodecanol, we examined the FT-IR spectra of W-VEGs prepared at different mixing ratios of EG to 1-dodecanol. The FT-IR spectra of W-VEGs shown in Figure 3a in a range of 4000 to 400 cm^−1^ clearly present IR bands of O-H stretching (~ 3450 cm^−1^), C-H stretching (3000 to 2800 cm^−1^), O-H/C-H bending (1800 to 1500 cm^−1^), C-O stretching (1239.2 cm^−1^), V = O/W = O stretching (1100 to 950 cm^−1^), and V-O/W-O stretching (700 to 400 cm^−1^). We found that the FT-IR spectrum of W-VEG prepared at the 1:1 mixing ratio of EG to 1-dodecanol was distinctly different from those of other W-VEGs. When we observed the C-H stretching bands in the range of 3000 to 2800 cm^−1^, the peak positions of W-VEGs prepared at 1:1 were around 2958.6, 2925.8, and 2853.5 cm^−1^ while those at 1:0, 1:0.5, and 1:2 appeared at around 2958.6, 2940.3, and 2879.5 cm^−1^, as shown in Figure 3b. The IR bands that appeared in the range of 1800 to 1500 cm^−1^ are ascribed to the overlapping spectra of O-H bending (~1670–1620 cm^−1^) and C-H bending in the EG units (~ 1598 cm^−1^) [25,26,27]. In Figure 3c, a shoulder peak in the range of 1790 to 1700 cm^−1^ appeared for W-VEG at the 1:1 mixing ratio, while only a strong band centered at around 1638 cm^−1^ was shown for all the other W-VEGs at the 1:0, 1:0.5, and 1:2 ratios. In addition, a strong IR band characteristic for C-O stretching appeared at 1239.2 cm^−1^ and a shoulder appeared at around 1286 cm^−1^ in the FT-IR spectrum of W-VEG at the 1:1 ratio. In contrast, two independent peaks at 1253.6 and 1231.5 cm^−1^ appeared in the FT-IR spectra of W-VEGs at the 1:0, 1:0.5, and 1:2 ratios. All these variations in the FT-IR spectrum of W-VEG at the 1:1 ratio strongly suggest that the dodecyloxy moiety combined with WOCl_4_ was incorporated into the W-VEG structure, thereby distinctly changing the peak positions of C-H stretching, O-H/C-H bending, and C-O stretching. One of the most plausible mechanisms to explain these results can be the reaction of NH_4_VO_3_ and EG with WOCl_4_ combined with both EG and 1-dodecanol to form W-VEG.

Then, we investigated the effects of the amount of tungsten precursor on the changes in the MIT temperature of W_x_VO_2_. We fixed the molar mixing ratio of EG to 1-dodecanol at 1:1 for further investigation. At the fixed conditions of NH_4_VO_3_, EG, and 1-dodecanol, the increasing concentrations of WOCl_4_ were used for the synthesis of W_x_VO_2_. All XRD patterns of W_x_VO_2_ synthesized at 0, 2.77, 5.53, and 11.08 mol% of WOCl_4_ to NH_4_VO_3_ in Figure 4a show that the VO_2_ (M) phase (JCPDS No. 43-1051) was successfully synthesized, although there exists a portion of the V_6_O_13_ phase with the addition of WOCl_4_. With the further addition of WOCl_4_ beyond 11.08 mol%, only V_6_O_13_ was synthesized (data not shown), indicating the oxidation of VO_2_ (M) due to the excessive addition of WOCl_4_. The MIT temperatures decreased from 68.07, 61.23, and 56.95 down to 51.12 °C at WOCl_4_ concentrations of 0, 2.77, 5.53, and 11.08 mol%, respectively, clearly showing the effects of tungsten doping (Figure 4b). This decrease in the MIT temperature is ascribed to the formation of the rutile phase VO_2_ (R) with the W doping, as indicated in literature [28]. The enhancement of (110) peak at 2θ = 26.8° is characteristic for the VO_2_ (R) phase [28], and clearly shown in the XRD patterns of W_x_VO_2_, supporting this mechanism. It should also be noted that the full width at half maximum (FWHM) of the melting transitions in the DSC thermograms (Figure 4b) drastically increased with the addition of WOCl_4_ from 3.9, 4.4, 6.2, to 11.4 °C. These results show that the crystallinity of W_x_VO_2_ was significantly decreased with increasing concentrations of WOCl_4_. Furthermore, the intensity of the melting peak drastically decreased with the increasing W doping, indicating that the activation energy for the MIT was reduced. In the monoclinic VO_2_ structure, V-V intervals are alternative (2.65 and 3.12 Å) while those in the rutile VO_2_ above the MIT are symmetric (2.87 Å). When V atoms are partially substituted with W atoms, the V-V intervals are shrunk, decreasing the structural differences between the monoclinic and rutile structure, and thereby reducing the activation energy [29]. The amounts of tungsten doped to vanadium in W_x_VO_2_ (M) were 0.33, 0.97, and 2.07 mol%, as determined by ICP-AES measurements (Table 1). These figures represent significantly decreased amounts in comparison to the tungsten precursor WOCl_4_. Hereafter, the calcinated W_x_VO_2_ from W-VEGs prepared at WOCl_4_ concentrations of 0, 2.77, 5.53, and 11.08 mol% are denoted by W_0.0033_VO_2_, W_0.0097_VO_2_, and W_0.0207_VO_2_, respectively. Our results showed that only 10~20% of tungsten added into the reaction mixtures was incorporated into the final W_x_VO_2_ compounds and indicate that a significant portion of WOCl_4_ that reacted with EG and 1-dodecanol did not participate in the formation of W-VEGs. We presumed that many of the species shown in Scheme 2 might not be appropriate for the formation of W-VEGs.

To examine the origination of these broadening melting peaks resulting from an increase in the WOCl_4_, we analyzed W-VEGs by measuring SEM images. Figure 5a shows that the VEG synthesized at the 1:1 mixing ratio of EG to 1-dodecanol without adding the WOCl_4_. The VEG has a nanowire morphology with a square or rectangular cross section. The nanowire morphology with a high aspect ratio over 10 is clearly different from that of VEG with a nanorod or microrod morphology prepared without using 1-dodecanol, which can be observed in Figure 2a. These results demonstrate that 1-dodecanol acts as a capping agent to block a specific crystal plane, thereby enhancing one-dimensional growth when VEG is synthesized. With the addition of WOCl_4_, the nanowire morphology disappears, presenting irregular crystal morphologies, as shown in Figure 5b–d. The crystal sizes of W-VEGs decreased from the micrometer-scale at 2.77 mol% to the nanometer-scale at 5.53 and 11.08 mol% of WOCl_4_, as shown in Figure 5b–d, respectively. We ascribe this disappearance of the nanowire morphology and the decrease in the crystal sizes to the increasing portion of dodecyloxy moieties in W-VEGs. When the bulky alkyl side chains are incorporated into the crystal structure of W-VEGs, they might act as defects to the formation of the one-dimensional chain-like structures shown in Scheme 1 and the crystallization of the chains.

W_x_VO_2_ compounds prepared via our modified polyol method utilizing 1-dodecanol present distinct thermochromic properties. We measured the transmittance spectra of the VO_2_ and W_0.0207_VO_2_ crystals dispersed in ethanol at 20 and 70 °C as two representatives. When the UV-Vis-NIR curves of 20 and 70°C were compared, there were no significant changes in the range of visible light shorter than 780 nm. However, in the NIR region, we observed that the transmittances of ethanol dispersions of VO_2_ and W_0.0207_VO_2_ at 70 °C were significantly decreased from those at 20 °C. The solar modulation *T_sol_* at each temperature was calculated by Equation (1), where *φ*(*λ*) and *Tr*(*λ*) are the solar irradiation spectrum for an air mass of 1.5 (corresponding to the sun standing 37° above the horizontal) [30] and transmittance at specific wavelengths *λ*, respectively, and Δ*T_sol_* was estimated by Equation (2) [9]. The luminous transmittance was also calculated by Equation (3) where *φ_lum_*(*λ*) is the spectral sensitivity of the light-adapted eye [18]. The estimated Δ*T_sol_* of VO_2_ without tungsten doping was 2.5% and that of W_0.0207_VO_2_ was 21.8%. The usefulness of our modified polyol method utilizing WOCl_4_ and 1-dodecanol to prepare tungsten-doped VO_2_ is evidenced by the significantly enhanced solar modulation of the ethanol dispersion of W_0.0207_VO_2_ with the manifestation of the tungsten doping. The luminous transmittance of W_0.0207_VO_2_ at 20 °C estimated by Equation (3) was 66.8% while that of VO_2_ was 78.2%. This decrease in the luminous transmittance with the tungsten doping can be ascribed to serious light scattering of W_0.0207_VO_2_ in comparison to VO_2_. Sizes of VO_2_ and W_0.0207_VO_2_ nanoparticles estimated by analyzing their SEM images shown in Figure 6c,d are 156 ± 81 and 287 ± 84 nm, respectively. These results indicate that the W_0.0207_VO_2_ nanoparticles with bigger sizes than the VO_2_ nanoparticles significantly scatter light, resulting in the decrease in transmittance. It should be noted that the average crystallite sizes of VO_2_ and V_6_O_13_ estimated by using XRD patterns in Figure 4a and the Scherrer equation [31] are 63.8 and 36.2 nm, respectively. The bigger size of the W_0.0207_VO_2_ nanoparticles than their crystallite sizes indicate that the nanoparticles have polycrystalline textures. On the other hand, the thermochromic properties gained in our study are compared with those of W_x_VO_2_ with the MIT temperature (*T_c_*) in a range of 40–45 °C in Table 2. The *T_lum_* at 20 °C and Δ*T_sol_* of W_0.0207_VO_2_ in this study are higher than those of reported materials except the *T_lum_* at 20 °C of the W-doped VO_2_/polyvinylpyrrolidone coating as presented in the Table 2 [32,33,34,35]. This comparison suggests that W_x_VO_2_ materials in our study have a great potential for thermochromic applications.
(1)$$T_{sol}\;=\;\frac{\int \varphi \left(\lambda \right)\cdot Tr\left(\lambda \right)\cdot d\lambda }{\int \varphi \left(\lambda \right)\cdot d\lambda }$$
(2)$$\Delta T_{sol}\;=\;T_{sol}\left(20\;{}^{\circ}C\right)-T_{sol}\left(70\;{}^{\circ}C\right)$$
(3)$$T_{lum}\;=\;\frac{\int \varphi_{lum}\left(\lambda \right)\cdot Tr\left(\lambda \right)\cdot d\lambda }{\int \varphi_{lum}\left(\lambda \right)\cdot d\lambda }$$

## 4. Conclusions

In this study, we carried out the synthesis of tungsten-doped VO_2_ via a polyol method based on ammonium metavanadate and EG by utilizing WOCl_4_ and 1-dodecanol as the tungsten source and capping agent, respectively. By conducting extensive characterizations using XRD, DSC, SEM, FT-IR, UV-Vis-NIR spectroscopy, and ICP-AES analysis, it was suggested that WOCl_4_ coupled with both EG and 1-dodecanol at an optimal ratio was responsible for the successful incorporation of tungsten into W-VEG in the creation of the calcinated compound, W_x_VO_2_. Other popular tungsten sources such as WCl_6_ and Na_2_WO_4_ were not effective for the tungsten doping in the polyol process, which indicates the usefulness of our modified polyol method that utilizes 1-dodecanol and WOCl_4_. However, it should be noted that the incorporation of bulky dodecyl chains into W-VEG seems to result in an increased FWHM value for the melting transition and that the current MIT temperature of W_x_VO_2_ at around 51 °C should be further decreased to near room temperature for practical thermochromic applications. In addition, the decrease in the luminous transmittance with the tungsten doping should be improved. These issues suggest that further investigations should be undertaken to find alcohol capping agents less bulky than 1-dodecanol and processes to avoid aggregations.

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
