# Peer review of "Synthesis of Tungsten-Doped Vanadium Dioxide Using a Modified Polyol Method Involving 1-Dodecanol"

_materials, 2020, doi:10.3390/ma13235384_

Round 1
Reviewer 1 Report
This paper describes an experimental study on the synthesis of tungsten-doped vanadium dioxide via a polyol process involving 1-dodecanol and the its characterization. Overall, their results are interesting, and the experiments are performed carefully. I recommend this paper for publication after some minor revisions.
- Introduction section.
Following sentences should be supported by the suitable references or the reviews. Please add the suitable references or the reviews in the text, respectively.
Page 1, line 28, “Smart windows~”
Page 1, line 30, “At 68 ~”
Page 1, line 33, “This metal-Insulator ~”
- Page 7, line 228, “The MIT temperatures decreased from ~”
Is this phenomenon usual? If not, this sentence would be better to explain more carefully. At least, it is recommended to mention a potential mechanism since this observation is one of the important results in the present study.
Author Response
Thanks a lot for your positive comments. We have sincerely responded to your comments to further improve the completeness of our manuscript. In the attachment, the blue-colored texts are our response and the red-colored texts are revised parts in our manuscript.

Reviewer 2 Report
In this manuscript entitled “Synthesis of Tungsten-Doped Vanadium Dioxide Using a Modified Polyol Method Involving 1-Dodecanol” the authors provided a comprehensive examination of a modified polyol method using 1-dodecanol as an additive to synthesize WxVO2(M). They investigate the influence of the mixing ratios of ethylene glycol (EG) and 1-dodecanol and the amounts of tungsten precursor on the synthesis of WxVO2 by using XRD, DSC, SEM, FT-IR, UV-Vis-NIR spectroscopy, and ICP-AES analysis. The tests provided are versatile, up-to-date, and complimentary and the results are well presented and give a good opportunity for further investigations.
I would like to recommend its publication in this journal after addressing the following recommendations:
- The sentence “Furthermore, the difference in the solar modulations at 70 °C was doubled to 20% in comparison to 11% at 20 °C.” in the abstract is not clear enough and does not give enough comparative quantitative information. The influence of mixing ratios of ethylene glycol (EG) and 1-dodecanol and the amounts of tungsten precursor on the morphology of WxVO2(M) crystals can be also added in the abstract.
- Except for SEM size measurements, the average grain size of both VO2 and V6O13 phase can be calculated according to the XRD spectra and additional conclusions can be drawn. Moreover, the change in the texture of the crystals is not commented on in the text.
- In Fig. 3b, c and d, please, indicate the particular bond stretching that is presented on a larger scale;
- The values of ICP-AES measurements of the samples should be tabulated and discussed in the text.
- More comments concerning the effect of grain sizes of the samples on both transmittance spectra of VO2 and W0.0207VO2 are needed.
Author Response

(The authors gave the same response as above.)

Reviewer 3 Report
The Article is devoted to the study of tungsten-doped VO2 (M), obtained via a modified polyol process with varying mixing ratios of EG to 1-dodecanol and WOCl4 concentrations. The Authors present the results of studies of W-VEGs by powder X-ray diffraction, differential scanning calorimetry, scanning electron microscopy, and infrared and transmittance spectroscopy. The obtained crystalline WxVO2 powders show a decrease in the metal-insulator phase transition temperature (MIT temperature) to 51 ° C and a temperature dependence of transmission. As the Authors point out, the 51 ° C value is too high for practical applications.
It should be noted that crystalline W-M: VO2 powders obtained by a similar method in [18] have significantly lower MIT temperatures. Unfortunately, the Article does not provide a comparison with the results of other works (for example [18]) in the same direction, and in the Conclusion of the Article, the Authors give only a list of necessary improvements without indicating how this can be done using their method.
The article will be of interest to Materials after taking into account the comments of the reviewer.
Author Response
Thanks a lot for your positive feedback. We have sincerely responded to comments and suggestions and hope that our revised version be complete enough for publication. In the below, the blue-colored texts are our response and the red-colored texts are revised parts in our manuscript.

Reviewer 4 Report
The authors have carried out a rigorous synthesis of tungsten-doped VO2 by a polyol method based on ammonium metavanadate and EG using WOCl4 and 1-dodecanol as the tungsten source and capping agent, respectively. Performing extensive characterizations using XRD, DSC, SEM, FT-IR,
UV-Vis-NIR spectroscopy and ICP-AES analysis. It should be noted the great utility of the modified polyol method used that uses 1-dodecanol and WOCl4.
Observations: the decrease in light transmittance with tungsten doping is a point that the authors should define better.
The current scientific bibliographic references should also be increased, enriching the bibliography and the state of the art.
It would be good to create a new discussion section where the findings are compared with those of other authors.
Author Response

(The authors gave the same response as above.)

Reviewer 5 Report
VO2 doped with W impurity is a well-known and effective method to reduce the MIT temperature of VO2. This work present a modified polyol method by using 1-dodecanol as an additive to synthesize W-doped VO2. I believe this work will be able to attract readers’ interest. However, in order to improve the quality of the paper, I have the following suggestions.
- English writing needs to be improved.
- In order to make it easier and clearer for readers to understand the synthesizing process of W-doped VO2, the authors should draw a schematic diagram of the synthesizing process and explain it in detail.
- I suggest that the authors should double confirm the MIT temperature of W-doped VO2 by variable temperature Raman, XRD and/or Resistance analysis, such as the following literatures.
https://doi.org/10.3390/coatings8120431
https://doi.org/10.3390/nano9081061
https://doi.org/10.3390/ma12132160
Author Response

(The authors gave the same response as above.)

Round 2
Reviewer 2 Report
The authors have carefully addressed all the review’s recommendations and the manuscript has been substantially improved.
Reviewer 4 Report
The article has been significantly improved.
Relevant scientific references have been added that have enriched the state of the art; still I give you a list of references that should be included in the article to improve the quality of the discussions.
Li, J., Liu, W., Zhang, X., Chu, P. K., Cheung, K. M. C., & Yeung, K. W. K. (2019). Temperature-responsive tungsten doped vanadium dioxide thin film starves bacteria to death. Materials Today, 22, 35–49. https://doi.org/10.1016/j.mattod.2018.04.005
Louloudakis, D., Vernardou, D., Spanakis, E., Suchea, M., Kenanakis, G., Pemble, M., … Kiriakidis, G. (2016). Atmospheric pressure chemical vapor deposition of amorphous tungsten doped vanadium dioxide for smart window applications. Advanced Materials Letters, 7(3), 192–196. https://doi.org/10.5185/amlett.2016.6024
Zhu, M. D., Shan, C., Li, C., Wang, H., Qi, H. J., Zhang, D. P., & Lv, W. Z. (2018). Thermochromic and femtosecond-laser-induced damage performance of tungsten-doped vanadium dioxide films prepared using an alloy target. Materials, 11(9). https://doi.org/10.3390/ma11091724
Ye, J., Zhou, L., Liu, F., Qi, J., Gong, W., Lin, Y., & Ning, G. (2010). Preparation, characterization and properties of thermochromic tungsten-doped vanadium dioxide by thermal reduction and annealing. Journal of Alloys and Compounds, 504(2), 503–507. https://doi.org/10.1016/j.jallcom.2010.05.152
González-Lezcano, R. A., & Del Río, J. M. (2015). Numerical analysis of the influence of the damping rings’ dimensions on interrupted dynamic tension experiment results. Journal of Strain Analysis for Engineering Design, 50(8), 594–613. https://doi.org/10.1177/0309324715601550
Reviewer 5 Report
The manuscript has been well revised, and the quality of the article has been significantly improved. Therefore, I recommend accepting the manuscript.